# Weight Control Registry Using Korean Medicine: A Protocol for a Prospective Registry Study

**DOI:** 10.3390/ijerph192113903

**Published:** 2022-10-26

**Authors:** Jiyun Cha, Eun Kyoung Ahn, Min-Ji Kim, So-Young Jung, Ho-Seok Kim, Eunjoo Kim, Hyun-Kyung Sung, Seon Mi Shin, Won-Seok Chung, Jun-Hwan Lee, Hojun Kim

**Affiliations:** 1Korea Institute of Oriental Medicine, Daejeon 34054, Korea; 2Department of Internal Korean Medicine, College of Korean Medicine, Daejeon University, Daejeon 34520, Korea; 3Nubebe Mibyeong Research Institute, Seoul 06634, Korea; 4Nubebe Korean Medical Clinic Bundang Center, Seongnam-si 13506, Korea; 5Department of Clinical Korean Medicine, Graduate School, Kyung Hee University, Seoul 02447, Korea; 6Department of Pediatrics, College of Korean Medicine, Semyung University, Jecheon-si 27136, Korea; 7Department of Internal Medicine, College of Korean Medicine, Semyung University, Jecheon-si 27136, Korea; 8Department of Korean Medicine Rehabilitation, College of Korean Medicine, Kyung Hee University, Seoul 02447, Korea; 9Korean Convergence Medical Science, KIOM School, University of Science & Technology (UST), Daejeon 34054, Korea; 10Department of Korean Medicine Rehabilitation, College of Korean Medicine, Dongguk University, Seoul 10326, Korea

**Keywords:** weight loss, obesity, nutrition, activity, sleep, depression, Korean medicine, herbal medicine, registry, real-world data

## Abstract

Lifestyle and physical characteristics affect body weight, and understanding these factors improves the precision of weight loss treatment. Many obese patients in Korea are receiving Korean medicine (KM) treatment, including herbal medicine and acupuncture, for weight loss. However, the real-world data (RWD) are insufficient in terms of being longitudinal and diverse. Weight Control Registry using KM is a prospective registry study that enrolls patients receiving KM treatment for weight loss and collects RWD from multiple clinics. The patients who are eligible for this study are aged 19–65 years, receive KM weight loss treatment, understand the study objectives, and consent voluntarily. Clinical data of patient characteristics and KM treatment patterns will be regularly collected until 2026. The longitudinal accumulation of various RWD will establish a high-quality study database for KM weight loss treatment. With this study, we expect to contribute to understanding the current trend of weight loss treatment with KM and solve further questions regarding this treatment.

## 1. Introduction

Obesity is a major risk factor for cardiovascular disease. It is associated with high healthcare burden and socioeconomic loss, and these factors deteriorate at a rate higher than that associated with other health conditions [1]. Asian populations tend to have 3–5% higher total body fat content [2], and a higher risk of cardiovascular disease and associated mortality than European populations with the same body mass index (BMI) [3,4]. Moreover, the prevalence of obesity and being overweight in Koreans exceeded 30% in 2005, reaching 33.8% in 2019. This increase is likely associated with lifestyle changes among Koreans due to globalization. The proportion of Korean adults performing aerobic exercise decreased by 10.5% in 2019 compared to that in 2014, and the proportion of Korean adults eating out at least once a day increased by 6.3% in 2019 compared to that in 2008 [5]. Evidence-based treatments for weight loss and comprehensive therapeutic approaches considering individual physical and lifestyle characteristics in primary care are required to deal with this change.

Many Korean patients who lose weight receive Korean medicine (KM) treatment, which may include herbal medicine (HM), acupuncture, electroacupuncture, and moxibustion. Several herbs traditionally used in Eastern Asia are preferred for weight loss treatment: Rhizoma coptidis, Ephedra sinica Stapf., Radix Lithospermi, Panax ginseng C. A. Mey, Carthamus tinctorius L, etc. In previous preclinical research, HM has shown beneficial anti-obesity effects and fewer adverse effects than chemicals. Potential anti-obesity mechanisms of herbs are decreasing proinflammatory cytokine expression in adipose tissues, increasing leptin sensitivity, reducing food intake, and alleviating insulin resistance [6,7]. Clinical effects of Korean HM and acupuncture for weight loss have been reported by systematic reviews and meta-analyses [8,9]. HM is usually added to lifestyle intervention and increases its effectiveness. According to a prior meta-analysis on the effectiveness of HM for weight loss, the clinical significance calculated from prior trials was defined as 2.5 kg or more weight difference between groups at the treatment endpoint [10]. From another meta-analysis, add-on therapy of HM and lifestyle intervention together achieved significantly lower body weight and BMI than that by placebo or Western medicine with the same lifestyle therapy [11]. Moreover, the multi-receptor targets of the HM formula, a compound of herbs, are effective for the holistic management of obese patients with various concomitant symptoms [12]. The clinical practice guidelines (CPGs) for KM treatment for obesity have been developed based on evidence from a prior randomized controlled trial [13]. However, the CPGs may not fully reflect the diversity and complexity of weight loss treatment by KM, which combines multiple therapies. Reviewing real-world data (RWD) from clinics can provide evidence for further studies, which would aid in understanding adherence rates, long-term prognoses, and precise treatment strategies. Nevertheless, high-quality RWD on weight loss treatment using KM remain insufficient.

The Weight Control Registry using KM (WECORE-KOM) is an initiative comprising multiple KM institutions. With the consortium of KM clinics and hospitals, this research registry will collect lifestyle and clinical data of patients receiving weight loss treatment from various KM institutions. This study is designed to understand lifestyle patterns of patients receiving KM for weight loss treatment and observe the long-term effects of KM on weight loss treatment. The resulting dataset will be provided to researchers for studies on KM weight loss treatment.

## 2. Methods

### 2.1. Objectives

The objectives of this study are as follows: (1) to develop a multicenter registry of KM weight loss treatment; (2) to collect RWD on KM weight loss treatment up to 5 years; (3) to describe the demographic, anthropometric, lifestyle, and clinical characteristics of patients receiving KM weight loss treatment; and (4) to contribute to solving future research questions on KM weight loss treatment, such as identifying prescription strategies by individual characteristics and evaluating long-term effects of KM weight loss treatment.

### 2.2. Study Design

The WECORE-KOM is a multicenter, longitudinal, and observational study that prospectively enrolls patients undergoing weight loss treatment with KM in Korea. Patients who provide informed consent to participate will receive personalized KM for weight loss treatment, without any pre-specified intervention. At baseline, data on the following variables will be collected: demographic and anthropometric characteristics, lifestyle data, medical/family history, device and laboratory test results, treatment preference and satisfaction, concomitant treatment, adverse events, and pattern identification, which is an analysis of disease status according to KM. The registry variable set for WECORE-KOM will include a recommended variable set developed by the National Agency for Korean Medicine Innovative Technologies Development (INKOM) [14]. Healthcare facilities in South Korea are classified into clinics with a single medical department or hospitals with multidisciplinary medical services. Patients with minor conditions can choose clinics with compact and prompt service. Those with chronic diseases or complications prefer hospitals to clinics in terms of KM medical service, since they provide a comprehensive service including medical diagnostic tests and conventional Western medicine. This study will include both types of KM institutions to observe clinical environment comprehensively and capture the differences between them. To facilitate the multicenter study, protocol and data collection status updates will be shared at regular meetings of the study staff. An electronic care report form (eCRF) will be created using the iCLICK system (http://ecrf.nikom.or.kr, accessed on 20 July 2022) and myTrial system version 2.0 for online surveys and automatic data capture (https://v2.mytrial.co.kr, accessed on 20 July 2022) by Bethesda Soft Co., Seoul, Korea, which was developed by INKOM for public eCRFs. The study flowchart is presented in Figure 1.

### 2.3. Study Participants

This registry aims to collect RWD from KM weight loss clinics. Patients receiving KM treatment for weight loss will be eligible for inclusion, regardless of their BMI.

The inclusion criteria will be as follows: (1) patients aged >19 and <65 years; (2) patients under KM weight loss treatment; and (3) patients who understand and voluntarily consent to the study protocol

The exclusion criteria will be as follows: (1) patients who refuse to participate; (2) patients disqualified by a KM doctor for weight loss treatment using KM; and (3) if a participant wishes to withdraw his or her participation, or if the patient is lost to follow-up (i.e., loss of contact for more than 1 year), his/her participation and data collection will be stopped.

### 2.4. Identification and Recruitment

Participants will be recruited from KM obesity clinics of five KM clinics/hospitals across Korea. KM doctors will treat patients who visit their clinics for weight loss and invite patients who meet the inclusion criteria to participate in this registry study. Researchers will fully explain the protocol to the patients, including the purpose of the study, data management processes, confidentiality, third-party data analysis protocols, and patient right to withdraw from participation. The patient will be requested to provide informed consent for participation. All the documents for this procedure were approved by the institutional review board (IRB) of each participating institute.

### 2.5. Follow-Up

A recommended data collection period is suggested for data quality. Data will be collected every 3 months (±2 weeks) for the first 6 months and every 6 months (±2 weeks) thereafter. Participants will be recommended to complete questionnaire-based assessments and undergo laboratory tests at the clinics. Participants unable to attend the designated follow-up period may be interviewed over the phone.

### 2.6. Data Collection, Variables, and Definitions

Data on the following variables will be collected to monitor the characteristics of KM weight loss treatment. Data collection details are presented in Table 1.

#### 2.6.1. Demographic Data

Data on age and sex will be collected for the basic demographic characteristics. Data on educational attainment, residential region, and occupation will be collected as factors associated with KM treatment preferences. Data on smoking status, drinking habits, and exercise routines will be collected as lifestyle factors that may affect body weight.

#### 2.6.2. Anthropometric Data

Data on the following variables will be measured and collected during an in-person visit: blood pressure, body temperature, and pulse rate (vital signs); weight, height, waist circumference, hip circumference, and waist-to-hip ratio (weight change); and BMI, body fat content, lean muscle mass, visceral fat level, obesity rate, and total body water content (body composition). If the participant does not visit according to the schedule, data on these parameters will be collected over the telephone to monitor treatment efficacy (weight gain rates).

#### 2.6.3. Lifestyle Data

Various lifestyle data will be collected using seven questionnaires. The International Physical Activity Questionnaire, which assesses the intensity of physical activity and sitting time, will be used to evaluate daily physical activity levels [15]. The Nutrition Quotient for adults will comprehensively evaluate meal quality and nutritional status from food intake and eating behavior [16]. The Korean version of the Beck Depression Inventory-II, which is the most used tool for screening depressive disorder, will be used to evaluate emotional disorders related to obesity and overweight [17]. The Pittsburgh Sleep Quality Index comprehensively evaluates sleep quality in seven related areas, including subjective sleep quality, sleep duration, and sleeping pill use [18]. The Epworth Sleepiness Scale assesses daytime sleepiness and hypersomnia by the degree of drowsiness in situations that may induce sleepiness [19]. The Fatigue Severity Scale evaluates physical vitality by the degree of fatigue felt in the past 7 days [20]. The Korean version of the Obesity-related Quality of Life scale, which is a self-reported questionnaire for six areas including psychosocial, physical, and daily health, will assess the impact of obesity on life [21].

#### 2.6.4. Medical/Family History

Past medical and family histories, including that of obesity, will be collected. If the participant has previously received any weight loss treatment, the relevant data will be collected.

#### 2.6.5. Pattern Identification of Korean Medicine

Pattern identification is an analysis of clinical data to determine the location, cause, and nature of disease [22]. It is an essential diagnostic method to KM clinical decision making, which identifies the physical condition and disease pattern of an individual based on symptoms. These patterns help to determine the prescription for HM or acupuncture and to personalize KM treatment ultimately. The participants will be instructed to complete the Obesity Pattern Identification Questionnaire, which is a tool to classify patients with obesity and those who are overweight into six subtypes of KM patterns [23]. The participants will also be instructed to complete other questionnaires that are used to identify the KM patterns (cold/heat pattern, deficiency/excess pattern, and Sasang constitutions) [24,25,26].

#### 2.6.6. Test using devices

Examining the tongue, pulse, and the face is a traditional diagnostic method used to determine the condition of the patient. Modern KM follows established protocols and uses dedicated devices to evaluate the tongue, pulse, and facial parameters associated with health and disease. Participants will undergo testing for tongue diagnosis using CTS-2000 (Daiseung Medics, Seoul, Korea), pulse diagnosis using DMP-life plus (Daeyo Medi, Seoul, Korea), and facial color analysis test using C922 (Logitec Inc., Lausanne, Switzerland).

#### 2.6.7. Laboratory Test

To evaluate any effects of long-term administration of HM on weight loss, liver and renal functions, and lipid and glucose levels, suitable laboratory test data will be collected at baseline and follow-up visits.

#### 2.6.8. Treatment

KM doctors will perform weight loss treatment and record its details in the eCRF. KM treatment type or frequency will have no restrictions, as all treatments will be individually determined based on patient characteristics. Data on weight loss treatment using KM and other concomitant treatments will be recorded, including INKOM-recommended variables. Drug adherence will be monitored to help determine treatment effectiveness.

#### 2.6.9. Motivation and Satisfaction

Motivation to visit will be enquired to understand the characteristics of patients who select KM treatment for weight loss. Patient satisfaction evaluation, which will be collected by an online survey, can account for patient willingness to continue/resume treatment, observed outcomes, overall experience, and costs.

#### 2.6.10. Adverse Events

Data on any adverse events will be collected to evaluate the safety of KM for weight loss treatment. Data on adverse events will include variables, such as type, severity, treatment specificity, and follow-up outcomes.

### 2.7. Quality Control

Data will be collected through an eCRF verified by the INKOM, and a data validation system will be used to prevent the input of out-of-range values. All data will directly be recorded in the eCRF. The clinical research coordinator (CRC) will give participants individual URLs to collect responses to the questionnaires via smartphones. The CRC of each institution will monitor all the electronic medical records, questionnaires, and measurement results from the devices and manage the supporting documents. Each case enrolled in the registry will be periodically reviewed manually by a clinical research associate, and data queries can be generated to provide feedback. If missing data or unresolved queries remain, the clinical research associate may notify each site to configure the standard operating procedure and help ensure that only the highest quality data are retained in the registry. Study monitoring will be conducted by clinical research associates affiliated with the Korea Institute of Oriental Medicine (KIOM), which is an institution separate from other medical institutions. This observational study does not involve a data monitoring committee because of the low risk of adverse events. The data collection platform of the WECORE-KOM is presented in Figure 2.

### 2.8. Sample Size

The purpose of this research registry is not to test specific hypotheses but accumulate RWD for further studies on KM weight loss treatment. Therefore, sample size calculations are not required for this study. However, considering the study period and the average number of patients presenting at the participating institutions, we plan to recruit at least 200 participants for a 5-year period.

### 2.9. Data Analysis

In principle, two-sided tests will be used for all statistical analyses, and the significance level will be set at 5%. Summary statistics on patient demographic, socioeconomic, and clinical characteristics will be presented per treatment type. Categorical data will be reported as frequencies and percentages. Continuous data will be reported as means and standard deviations. Subgroup analyses will be performed after testing the normality of distribution assumption. Between-group comparisons of continuous variables will be performed using the independent *t*-test or Wilcoxon rank-sum test. Comparisons of three or more groups will be performed with analysis of variance (normally distributed data) and the Kruskal–Wallis test (non-normally distributed data). Categorical variables will be compared using the chi-squared or Fisher’s exact test. For missing values, the last observation carried forward method will be used. All analyses will be performed using SPSS version 24 (IBM Corp., Armonk, NY, USA) or R version 4.1.0 (The R Foundation for Statistical Computing, Austria). Additional IRB approvals will be requested, as required, to test study hypotheses in the future.

### 2.10. Governance, Oversight, and Data Sharing

Data will be stored safely in a dedicated collection server (http://ecrf.nikom.or.kr, accessed on 20 July 2022) operated by the INKOM under the Korea Ministry of Health and Welfare. The server provides a cloud-based solution for clinical trials and grants limited data access rights. Data input will be possible only by clinical researchers, including the CRC, and only the designated data manager will be allowed to extract data. Registry data analysis will be possible both during and after the data collection period. Nevertheless, the resulting datasets will be provided only to individuals or team researchers who have received IRB approval for suitable research purposes. In such cases, only data on relevant variables will be extracted. For the duration of the registry operation, the KIOM data manager will oversee the database, including providing access rights, and INKOM will be ultimately responsible for the registry administration. An initial report will be generated 1–2 years after registry launch, enabling early data review and clinical research scope assessment.

### 2.11. Ethical Approval and Trial Registration

The following IRBs approved this study: Dongguk University Ilsan Oriental Hospital (DUIOH 2021-10-009-002); Semyung University Korean Medicine Hospital, Chungju (SMCJH 2111-13); Semyung University Korean Medicine Hospital, Jecheon (SMJOH-2021-12); Kyung Hee University Korean Medicine Hospital (KOMCIRB 2021-11-004-001); and Kyung Hee University (KHSIRB-21-551). This study was registered at the Clinical Research Information Service (cris.nih.go.kr) on 17 March 2022 (No. KCT0007089). Recruitment started on 18 April 2022, at Semyung University Oriental Medicine Hospital and will be completed on 31 December 2026.

## 3. Discussion

We will collect various clinical data from those who receive KM for weight loss treatment. Anthropometric and laboratory test data gained from RWD will contribute to verify effectiveness and safety of KM for weight loss treatment. Long-term observation of weight will help to predict weight regain that tends to occur within the first 5 years after weight loss. Frequency and results of anthropometric measurements are associated with successful weight control [27]. It is reported that HM complex may be safe for the treatment of obesity compared to conventional medicine or placebos [8]. Laboratory test data will be observed to support the safety of HM in obese patients. Lifestyle changes due to weight loss is another area of concern. Some lifestyle patterns commonly found in obese or overweight patients are associated with weight gain. High calorie intake and low physical activity are main mechanisms of weight gain, and it is essential to observe dietary and physical behaviors of obese patients [28]. In terms of psychology, certain types of depression and obesity may be risk factors for each other [29]. Sleep disorder and fatigue are strongly associated with weight gain. Obesity contributes to poor sleep quality and is associated with hyposomnia and hypersomnia. Endocrine and metabolic changes induced by sleep disorders can contribute to being overweight. High visceral fat in obese patients is associated with increased plasma inflammatory cytokine levels and the risk of sleep apnea, and this causes low sleep quality, high daytime sleepiness, and high risk of cardiovascular diseases [30,31]. Although KM weight loss treatment is widespread in Korea, correlations between lifestyle change by weight loss and the effectiveness of KM therapies are still unclear. Observing various lifestyle factors during weight loss, we expect to discover the correlation among lifestyle behaviors, weight, and KM treatment.

In KM treatment, patient patterns based on everyday health lifestyle are emphasized for precise prescription of the HM. This is called “pattern” or “constitution” differentiation in traditional Eastern Asian medicine (TEAM). It is essential to identify the diverse health information and detailed prescriptions to analyze the prescription patterns of Korean HM. However, a claims database without information on KM diagnostic health or non-reimbursable treatment, which accounts for a large portion of KM treatment, creates an evidence gap between KM claims data and RWD. Establishing an RWD registry, which would allow researchers to collect data under protocol, is a method to collect patient data to fill this gap and increase data usability. Moreover, reviewing RWD often becomes an effective alternative plan for conventional clinical trials. Single-center or single-treatment clinical studies fail to capture the complexity of uncontrolled clinical practice. CPGs, based on clinical trials, rarely account for the diversity of the real world with characteristics, preferences, comorbidities, and sensitivities of individuals. This often limits the application of CPGs to the real world and complicates clinical decision making. Reviewing RWD may extensively capture clinical information, adverse events, and prognoses by individual characteristics, which can be missed in conventional trials. In TEAM, registry research for collecting RWD of various diseases has been recently conducted [32,33,34]. This study has some differences from a previous study suggesting a KM registry of HM for weight loss [35]. This study will collect comprehensive patient information, including physical data and KM patterns, as well as lifestyle questionnaires. Clinics and hospitals of KM will participate, and information on all KM therapies, including HM and acupuncture, will be collected to incorporate clinical diversity and complexity as much as possible.

General limitations of registry studies should be considered in interpreting the results. In registry studies without randomization, selection imbalance of subjects and therapies may cause bias. Preference of physicians in certain diagnoses or therapies and characteristics of actively participating subjects may confound the results. Researchers should observe these fully and pay close attention to topic selection and result interpretation. Nevertheless, with a review of real-world clinical practice from multiple centers, WECORE-KOM will contribute to understanding KM treatment for weight loss and improve weight loss intervention in Korean primary care.

## 4. Conclusions

Personalization according to individual habits and physical characteristics is considerably important in weight control treatment. Although comprehensive clinical data analysis is required for this, high-quality clinical data on KM weight loss treatment remain insufficient. This long-term RWD study will accumulate clinical evidence combined with lifestyle characteristics for weight loss treatment using KM, based on real-world practice data. We expect that WECORE-KOM will provide significant knowledge about the efficacy, safety, and personalized strategy of KM weight loss treatment.

## Figures and Tables

**Figure 1 ijerph-19-13903-f001:**
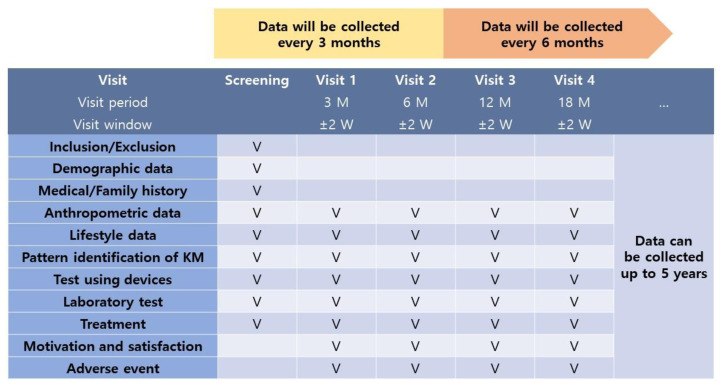
Study flowchart. KM, Korean medicine.

**Figure 2 ijerph-19-13903-f002:**
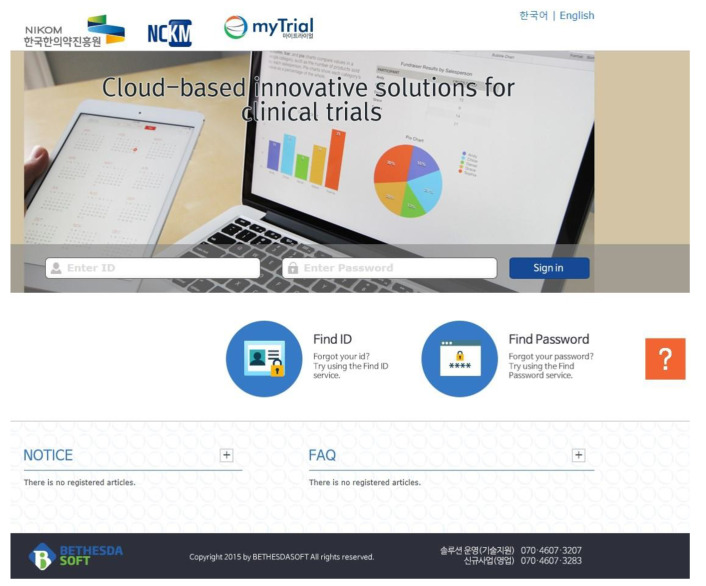
Data collection platform of the WECORE-KOM. The main page of the electronic data capture system is operated by the INKOM under the Ministry of Health and Welfare of Korea (accessed 20 July 2022). INKOM, National Agency for Korean Medicine Innovative Technologies Development; WECORE-KOM, Weight Control Registry using Korean Medicine.

**Table 1 ijerph-19-13903-t001:** Detailed data variables.

Domain	Items
Demographic data	Age, sex, educational attainment, region, occupation, smoking status, drinking habits, exercise routine
Anthropometric data	Vital signs, body weight, height, BMI ^1^, body fat, body muscle, visceral fat level, obesity rate, total body water content
Lifestyle data	iPAQ ^2^, NQ ^3^, BDI-II ^4^, PSQI-K ^5^, ESS ^6^, FSS ^7^, KOQOL ^8^
Medical/family history	History of obesity
Pattern identification of KM ^9^	OPIQ ^10^, cold/heat pattern, deficiency/excess pattern, Sasang constitutions
Test using devices	Tongue, pulse, facial color patterns measured by devices
Laboratory test	Liver function, renal function, lipid and glucose level tests
Treatment	Herbal medicine, acupuncture, moxibustion, cupping, physical therapy, others
Motivation and satisfaction	Motivation to visit, willingness to revisit for weight loss, overall treatment satisfaction
Adverse event	Adverse events during KM ^9^ treatment

^1^ BMI, body mass index; ^2^ iPAQ, International Physical Activity Questionnaire; ^3^ NQ, Nutrition Quotient for adults; ^4^ BDI-II, Beck Depression Inventory-II; ^5^ PSQI-K, Pittsburgh Sleep Quality Index; ^6^ ESS, Epworth Sleepiness Scale; ^7^ FSS, Fatigue Severity Scale; ^8^ KOQOL, Korean version of the Obesity-related Quality of Life scale; ^9^ KM: Korean medicine; ^10^ OPIQ, Obesity Pattern Identification Questionnaire.

## Data Availability

Not applicable.

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
