# Peer review of "Weight Control Registry Using Korean Medicine: A Protocol for a Prospective Registry Study"

_ijerph, 2022, doi:10.3390/ijerph192113903_

Round 1

Reviewer 1 Report

The authors have provided a study protocol for the Weight Control Registry using KM (WECORE-KOM). This is a multi-institution effort to develop a weight loss registry for Korean patients interested in weight loss who receive Korean medicine (KM). The purpose of the study is to gather real-world data to better understand weight loss treatment using KM. There are a few items the authors could address to further strengthen this manuscript.

The authors made a number of references to lifestyle changes related to weight management. Consider additional information in the introduction to help the reader understand the roll of KM in relation to lifestyle approaches to weight management. Are these approaches mutually exclusive? Would somebody use both?

In section 2.1 remove the bullet points and use sentences.

In section 2.2 the authors differentiate classification of clinics and hospitals. Help the reader understand the difference.

In section 2.3 either create a table of criteria or use sentences. In either case, remove the bullet point format.

In section 2.6.2 consider “and collected at an in-person visit;”

Author Response

Response to Reviewer 1 Comments

Point 1: The authors made a number of references to lifestyle changes related to weight management. Consider additional information in the INTRODUCTION to help the reader understand the roll of KM in relation to lifestyle approaches to weight management. Are these approaches mutually exclusive? Would somebody use both?

Response 1:

Korean herbal medicine (HM) for weight loss can assist lifestyle intervention, especially dietary therapy. Experimental and empirical evidences have revealed that HM contributes to reducing food intake and suppressing the appetite. For instance, Coicis Semen, which is a common ingredient in the HM formula for weight loss, reduces leptin and TNF-α messenger RNA expression in the white adipose tissues resulting in loss of appetite and weight in diet-induced obese rats.(1) Moreover, the multireceptor targets of the HM formula, a compound of herbs, can be effective for the holistic management of obese patients with various symptoms throughout their body.(2)

HM is usually added on to lifestyle intervention and it increases its effectiveness. According to a meta-analysis by Wong et al., add-on therapy of HM and lifestyle intervention together achieved significantly lower body weight and BMI than that by placebo or Western medicine with the same lifestyle therapy.(3)

Thank you for your suggestion. We have added more information about HM for weight loss in the Introduction section as follows:

Page 2, Line 67-73:

" HM is usually added on to lifestyle intervention and it increases its effectiveness. According to a prior meta-analysis, add-on therapy of HM and lifestyle intervention together achieved significantly lower body weight and BMI than that by placebo or Western medicine with the same lifestyle therapy. (3) Moreover, the multireceptor targets of the HM formula, a compound of herbs, are effective for the holistic management of obese patients with various concomitant symptoms. (2)”

(1) Kim, S.O.; Yun, S.J.; Jung, B.; Lee, E.H.; Hahm, D.H.; Shim, I.; Lee, H.J. Hypolipidemic effects of crude extract of adlay seed in obesity rat fed high fat diet: relations of TNF-α and leptin mRNA expressions and serum lipid levels. Life Sci 2004, 75, 1391-1404.

(2) Zhang, G.B.; Li, Q.Y.; Chen, Q.L.; Su, S.B. Network pharmacology: a new approach for Chinese herbal medicine research. Evid based complementary altern med 2013, 2013.

(3) Wong, A.R.; Yang, A.W.H.; Li, K.; Gill, H.; Li, M.; Lenon, G.B. Chinese herbal medicine for weight management: a systematic review and meta-analyses of randomised controlled trials. J Obes 2021, 2021.

Point 2: In section 2.1 remove the bullet points and use sentences.

Response 2:

Thank you for the suggestion. I have rearranged this paragraph into sentences as per your recommendation. (Page 2, Line 92 – Page 3, Line 98)

Point 3: In section 2.2 the authors differentiate classification of clinics and hospitals. Help the reader understand the difference.

Response 3:

In the medical system of South Korea, clinics and hospitals are classified according to the size of inpatient services. Generally, clinics offer medical services of a single department, primarily for outpatients with minor conditions. On the other hand, hospitals provide medical services that encompass both inpatient and outpatient with medical staff from various departments (e.g., collaboration between Korean medicine doctors and conventional Western medicine doctors). Outpatients with major conditions or complications tend to prefer hospitals to clinics.

To incorporate your suggestion, we have revised the sentence as below:

Page 3, Line 109-114:

"Health care facilities in South Korea are classified into clinics with a single medical department or hospitals with multidisciplinary medical services. Patients with minor conditions can choose clinics with compact and prompt service. Those with chronic diseases or complications prefer hospitals to clinics in terms of KM medical service, since they provide a comprehensive service including medical diagnostic tests of conventional Western medicine."

Point 4: In section 2.3 either create a table of criteria or use sentences. In either case, remove the bullet point format.

Response 4:

Thank you for the suggestion. I have rearranged this paragraph into sentences, as recommended by you. (Page 4, Line 130-138)

Point 5: In section 2.6.2 consider “and collected at an in-person visit;”

Response 5:

Thank you for the suggestion. I have revised the sentence as per your recommendations. (Page 5, Line 178)

Reviewer 2 Report

The authors described the protocol of a weight control registry using Korean medicine via five clinics and hospitals. The protocol is fairly clear to understand, but I have a few minor comments on introduction and methods section.

1. Can the authors add more information about previous Korean medicine registries if they existed?

2. What was the effect size of Korean medicine on obesity or other diseases/symptoms in literature?

3. Can the authors clarify the meaning of "solving research questions on the prescription patterns by individual?" (line 91) what are the research questions? 

4. Percent would be a better word than ratio (line 274).

5. How do the participants respond to eCRF? Using a tablet, phone, or computer? Will the research staff assist the old participants unfamiliar with the device?

Author Response

Point 1: Can the authors add more information about previous Korean medicine registries if they existed?

Response 1:

In Korean medicine and traditional Chinese medicine (usually referred to as traditional Eastern Asian medicine, TEAM), registry researches for collecting real-world data of various diseases, such as migraine, rhinitis and cancer, have been conducted recently. (1-3)

We have added the following information in the manuscript:

Page 9, Line 359-360:

“In TEAM, registry researches for collecting RWD of various diseases have been recently conducted.” (1-3)

(1) Lyu, S.; Zhang, C.S.; Zhang, A.L.; Sun, J.; Xue, C.C.; Guo, X. Migraine patients visiting Chinese medicine hospital: Protocol for a prospective, registry-based, real-world observational cohort study. PloS one 2022, 17(3), e0265137.

(2) Chu, H.; Jang, B.H.; Lee, E.; Moon, S.; Ham-soa Clinic KM doctors group. Combined Korean medicine therapies in children with allergic rhinitis: a multi-center, observational explanatory registry trial: a study protocol. Medicine 2021, 100, e28181. doi:10.1097/MD.0000000000028181

(3) Bae, K.; Kim, E.; Choi, J.J.; Kim, M.K.; Yoo, H.S. The effectiveness of anticancer traditional Korean medicine treatment on the survival in patients with lung, breast, gastric, colorectal, hepatic, uterine, or ovarian cancer: a prospective cohort study protocol. Medicine 2018, 97, e12444. doi:10.1097/MD.0000000000012444

Point 2: What was the effect size of Korean medicine on obesity or other diseases/symptoms in literature?

Response 2:

Thank you for the comment.

According to a prior meta-analysis on the effectiveness of herbal medicines for weight loss, the clinical significance calculated from prior RCT studies was defined as 2.5 kg or more weight difference between groups at treatment endpoint.(1)

However, it is difficult to define the appropriate effect size in this study. This is a registry collecting real-world data and it can be extended to various studies: the effectiveness of herbal medicines for weight loss, usability evaluation, user satisfaction evaluation, current status analysis of weight loss related factors, or any subject the researchers want. Effect size or sample size calculation is not applicable in a registry study without a pre-determined hypotheses to verify.(2) Researchers analyzing our data in the future have to formulate their research hypothesis and consider the appropriate effect size by it.

(1) Maunder, A.; Bessell, E.; Lauche, R.; Adams, J.; Sainsbury, A.; Fuller, N.R. Effectiveness of herbal medicines for weight loss: A systematic review and meta‐analysis of randomized controlled trials. Diabetes Obes Metab 2020, 22(6), 891-903.

(2) Gliklich, R.E.; Dreyer, N.A.; Leavy, M.B. (Eds.). Registries for evaluating patient outcomes: a user’s guide. 2014.

Point 3: Can the authors clarify the meaning of "solving research questions on the prescription patterns by individual?" (line 91) what are the research questions?

Response 3:

This study aims to collect real world data of KM weight loss treatment prospectively, not to examine specific hypotheses directly.

However, considering the collecting variables of this study, it is anticipated to contribute to findings on the following subjects: Identifying HM prescription strategies according to individual characteristics, such as anthropometric information, comorbidities, concomitant symptoms, and KM patterns and evaluating long-term weight maintenance of KM weight loss treatment.

Based on your comment, we have described the research objectives in detail as follows:

Page 2, Line 95 – Page 3, Line 96-98:

“(4) to contribute to solving future research questions on KM weight loss treatment: for instance, identifying prescription strategies by individual characteristics and evaluating long-term effects of KM weight loss treatment.”

Point 4: Percent would be a better word than ratio (line 274).

Response 4:

Thank you for the suggestion. I have made a revision based on your recommendation. (Page 7, Line 284)

Point 5: How do the participants respond to eCRF? Using a tablet, phone, or computer? Will the research staff assist the old participants unfamiliar with the device?

Response 5:

Thank you for the comment.

For answering the questionnaires, participants do not have direct access to the eCRF, but connect to the web survey link sent to their smartphone. Participants' answer data will be automatically recorded in the eCRF. The CRCs explain this process to participants and help them to conduct the web survey on electronic devices. The CRCs are also responsible for completing eCRF and they check the accuracy of recorded data.

Considering your comment, we have made the following modification to the statement to clarify the meaning:

Page 6, Line 252-255:

“All data will directly be recorded in the eCRF. The clinical research coordinator (CRC) of each institution will monitor all the electronic medical records, questionnaires, and measurement results from the devices and manage the supporting documents."

Round 2

Reviewer 2 Report

Thank you for all the responses. I strongly suggest including your responses for #2 ("According to a prior meta-analysis on the effectiveness of herbal medicines for weight loss, the clinical significance calculated from prior RCT studies was defined as 2.5 kg or more weight difference between groups at treatment endpoint.(1)") and 5 (e.g., giving individual URLs to collect responses to the questionnaires via smartphones) in the manuscript on page 2, lines 68-73, and page 6, lines 252-255.

Author Response

Response to Reviewer 2 Comment

Point 1: Thank you for all the responses. I strongly suggest including your responses for #2 ("According to a prior meta-analysis on the effectiveness of herbal medicines for weight loss, the clinical significance calculated from prior RCT studies was defined as 2.5 kg or more weight difference between groups at treatment endpoint.(1)") and 5 (e.g., giving individual URLs to collect responses to the questionnaires via smartphones) in the manuscript on page 2, lines 68-73, and page 6, lines 252-255.

Response 1:

I appreciate for your kind comments.

Based on your comments, I've added response contents to the manuscript (Page 2, Lines 68-71; Page 6, Lines 255-256).
